# Epigenetic Inheritance: Intergenerational Effects of Pesticides and Other Endocrine Disruptors on Cancer Development

**DOI:** 10.3390/ijms23094671

**Published:** 2022-04-23

**Authors:** Heloiza Diniz Nicolella, Sonia de Assis

**Affiliations:** 1Georgetown University Medical Center, Washington, DC 20057, USA; hn258@georgetown.edu; 2Lombardi Comprehensive Cancer Center, Washington, DC 20057, USA

**Keywords:** pesticide, environment, epigenetic inheritance, intergenerational transmission, pediatric and adult cancer

## Abstract

Parental environmental experiences affect disease susceptibility in the progeny through epigenetic inheritance. Pesticides are substances or mixtures of chemicals—some of which are persistent environmental pollutants—that are used to control pests. This review explores the evidence linking parental exposure to pesticides and endocrine disruptors to intergenerational and transgenerational susceptibility of cancer in population studies and animal models. We also discuss the impact of pesticides and other endocrine disruptors on the germline epigenome as well as the emerging evidence for how epigenetic information is transmitted between generations. Finally, we discuss the importance of this mode of inheritance in the context of cancer prevention and the challenges ahead.

## 1. Introduction

According to the World Health Organization (WHO) (2013) [1], a pesticide is any substance or mixture of substances of chemical or biological ingredients, intended for repelling, destroying or controlling any pest, or regulating plant growth. Pesticides are widely used in agriculture and while pest management is critical to food production, the risk it poses to human health and the environment is evident, especially in the long term [2].

During the production and application of pesticides, professionals can be exposed to the chemical agent. However, the general population is also susceptible to this exposure, due to contamination of the soil, water and food, as well as the biological accumulation trough the food chain, since the pesticides have cumulative properties and circulate in ecosystems [3,4].

The absorption of pesticides in the body can occur in different ways, depending on the species, metabolic characteristics and susceptibility to the toxicant [4]. These factors also affect the body’s response to the toxic agent, resulting in more or less severe effects. Exposure to pesticides mixtures also occurs and triggers synergistic toxicity [5]. There may be an interaction between different pesticides in the body, altering their toxicokinetics, as one pesticide can affect the absorption rate of another by binding to the efflux membrane transporter protein and altering the distribution of the pesticide to its molecular target [6].

Evidence of the effects of endocrine disruptors in humans is still an issue, because manifestations become evident many years later. In addition, people are exposed to a variety of chemical substances with different mechanisms of action throughout their lives, and the interactions between them can cause different consequences from the isolated compounds [7]. However, several studies demonstrate the pleiotropic impact of endocrine disruptors, including pesticides, on human health. These chemicals can cause neurological and immunological toxicity, as well as carcinogenesis in animals and humans. Exposure to pesticides also increases the risk of other chronic diseases such as diabetes and heart problems. These compounds can also harm offspring through in utero exposures and lead to congenital abnormalities, in addition to affecting growth and normal development [7,8].

Furthermore, studies in both animal models and human cohorts have linked exposure to pesticides and other environmental toxicants to defects in the parental germline. For instance, in utero exposure to DDE, a DDT metabolite, impairs male fertility and causes epigenetic alterations in the male germ-line in rats [9]. Other pesticides have similar effects: prenatal dioxin induces sperm epimutations and disease in subsequent generations [10]. Studies in human cohorts support the findings in animal models and report an association between pesticide exposure, including DDT, and sperm quality (reviewed in [11]). The female germline and reproductive health have also been shown to be sensitive to the endocrine disrupting action of pesticides, with long-term adverse effects to offspring [12]. Accordingly, epidemiologic studies have linked parental occupation—and associated pesticide exposure—to cancer and other diseases in their children [13,14,15,16].

Given their impact on the germline, pesticides and other endocrine disruptors have been linked to transgenerational epigenetic inheritance. Epigenetic patterns are faithfully transmitted between cells during mitotic division, but accumulating evidence suggests that epigenetic information can also be transmitted between generations [17,18]. Accordingly, studies suggest that the progeny’s health phenotypes can be acquired in an epigenetically inherited manner due to parental environmental exposures [18,19]. This environmentally induced disease risk has been shown to be transmitted to the offspring via epigenetic mechanisms through both the female and male germ-lines [20,21,22]. Although most of the evidence for this mode of disease inheritance comes from maternal exposures in pregnancy, we [23,24,25,26] and others [27,28,29] have shown that paternal exposures in the pre-conception window are also important in determining disease outcomes in the offspring.

This review summarizes the evidence on parental exposures to pesticides and the predisposition to cancer development in the next generation in both human populations and animal studies. We also explore how exposure to pesticides can influence intergenerational disease susceptibility via germline epigenetic alterations.

## 2. Pesticides

Pesticides—which are classified according to the target organism—include insecticides, herbicides, fungicides or fumigants. They are further subclassified, based on their active principles, as organochlorines, organophosphates, carbamides, pyrethroids, and others [30].

### 2.1. Organochlorines

Organochlorines are compounds that contain at least one covalently bonded chlorine atom which may present a wide structural diversity and, consequently, different chemical properties [31]. These pesticides were initially used to control typhus and malaria and later were used in food crops. However, because they are persistent organic pollutants (POPs) and cause environmental damage, they were banned by several countries or strictly controlled in those countries that still allow their application [32]. POPs are considered silent killers due to their resistance to natural degradation, which allows them to bio-accumulate and persist in the environment. Most POPs are highly soluble in lipids and accumulate in living organisms, including humans [33].

Exposure to organochlorines can occur through different routes including contact with the skin, inhalation and ingestion. The main route of exposure to the compounds occurs through the ingestion of contaminated food, especially with a higher fat content. Additionally, the consumption of contaminated breast milk is another important means of exposure [34].

Organochlorines are also able to cross the placental barrier, causing fetal abnormalities, as well as physiological changes that may manifest in the long term. In addition to causing endocrine-disruption [35,36], their accumulation in the adipose tissue is related to the development of physiological disorders and becomes an indication of the presence of the substance in the human body [37].

Dichlorodiphenyltrichloroethane (DDT) is one of the main organochlorine pesticides. Although its use was banned in several countries in the 1970s, the WHO authorized its use in countries with a high incidence of malaria cases in 2006 to reduce mortality rates [38]. While the use of DDT has been restricted in the United States (U.S.). and Europe for decades, DDT has a prolonged half-life and is lipophilic, bio-accumulating in the food chain, and can still be detected in fatty tissues and in the circulation of animals and humans [39]. In the U.S., the highest concentrations of DDE (dichlorodiphenyldichloroethylene), DDT’s main metabolite [34], are found in minority populations and recent immigrants [40,41].

Because DDT and its metabolites are endocrine disruptors, acting on the estrogen and androgen pathways, they cause adverse effects on the reproductive system. Reproductive toxicity depends on the affected species, as well as on the dose and exposure conditions. Additionally, studies demonstrate hepatotoxic and carcinogenic effects of this pesticide [38,42]. Intrauterine exposure to DDT has been associated with increased cancer risk in the offspring [43]. Overall, the data in published literature demonstrate that the effects of this class of pesticide are highly deleterious to human health, with long-term consequences.

### 2.2. Organophosphates and Carbamates

In addition to organochlorines, other pesticides are also widely used in pest control and likewise cause damage to the environment and human health. Organophosphates are a highly toxic class of insecticide, the most common of which is glyphosate. It is estimated that these toxicants are responsible for about 100,000 deaths a year. These compounds are able to phosphorylate several enzymes and proteins, but its toxicity is linked to the inhibition of the acetylcholinesterase enzyme [44,45]. Exposure to organophosphates can induce long-term adverse effects through a non-cholinergic process. Mothers exposed to organophosphate can transfer it to the fetus via the placenta or amniotic fluid, significantly affecting the child’s development [46].

In addition to congenital abnormalities, cognitive and neurobehavioral deficiencies, exposure to organophosphates may favor the development of cancer [47]. Exposure to this type of pesticide was associated with an increased risk of breast cancer, and maternal exposure was also associated with the development of acute lymphoblastic leukemia in childhood [48,49]. It has been proposed that maternal genotoxic exposures could trigger nonhomologous chromosome rearrangement and initiate cancer [48,49].

Carbamates, a class of pesticides that includes aldicarb and carbofuran, also have an endocrine disrupting activity. These pesticides also exert their toxicity by inhibiting acetylcholinesterase. However, the enzyme is reactivated more quickly compared to organophosphates exposure. Thus, carbamates poisoning is less severe and chronic side effects are rare. However, this toxicant can cause reproductive toxicity, including endocrine disruption and infertility [45,50]. In vivo studies have shown that prenatal exposure to carbamates can affect offspring development, as well as contributing to a significantly higher tumor incidence in the first generation whose parents were exposed to the pesticide [51,52].

Other classes of chemical pesticides including pyrethroids, triazines, and neonicotinoids have also been shown to potentially cause endocrine disruption and reproductive effects in animals [53].

## 3. Parental Exposure to Pesticide and Endocrine Disruptors and Cancer Predisposition in the Next Generation

The developmental origins of health and disease (DOHAD) hypothesis posits that increased risk of disease in adulthood can result from an adverse developmental environment that reprograms cellular and tissue responses to normal physiological signals in a manner that increases disease susceptibility later in life [54,55]. Consistent with that, prenatal exposure to environmental pesticides and other endocrine disruptors can promote dysregulation of the fetal epigenome, leading to pathologies such as cancer during childhood or adulthood and even transgenerationally [56]. While gene-environment interactions can be important in determining cancer risk in the context of familial syndromes [57], it is becoming evident that pre-conception exposures can lead to germline epigenetic changes and increase cancer susceptibility in the next generation [23,24,25,26]. In this section, we review that evidence linking pre-conception and prenatal exposure to pesticides and other endocrine disruptors and cancer in the progeny.

### 3.1. Pediatric Cancers

#### 3.1.1. Leukemia

Epidemiological studies have shown that exposure to pesticides, even at low doses, during pregnancy or early childhood increases the risk of leukemia in children [58,59]. Chen et al. [60] carried out a meta-analysis that indicated that childhood exposure to indoor, but not outdoor, residential pesticides significantly increases the risk of childhood leukemia as well as lymphomas.

A meta-analysis of several epidemiologic studies found a relationship between prenatal maternal exposure to pesticides and the development of childhood leukemia. An association with paternal occupational exposure to pesticides was also identified, but findings were less consistent [61]. However, paternal (peri-conception) occupational pesticide exposure was associated with the risk of acute lymphoblastic leukemia (ALL) in the offspring in another human cohort. Results were more evident in children diagnosed at 5 years of age or older and in those diagnosed with T cell ALL [14].

A significantly higher incidence of t(8;21) translocation in umbilical cord blood samples of newborns from mothers exposed to pesticides during prenatal care was reported [62]. These results reinforce the relationship between prenatal pesticide exposures and the leukemic-associated t(8;21) (q22;q22) translocation, one of the most common cytogenetic abnormalities in childhood acute myeloid leukemia (AML). In line with the above study, Lu and coworkers [63] showed that pesticide chlorpyrifos induced mixed-lineage leukemia (MLL) translocations through caspase 3-dependent genomic instability and topoisomerase II inhibition in human fetal liver hematopoietic stem cells (HSC). The authors proposed that MLL abnormalities in the appropriate HSC cells may be the molecular basis for infant leukemia.

#### 3.1.2. Hodgkin and Non-Hodgkin Lymphoma

An increased risk of non-Hodgkin lymphoma (NHL) in children has also been associated with parental exposure to pesticides. A significant association between the risk of NHL and the increased frequency of reported pesticide use in the home has been reported. Maternal exposure to household insecticides during pregnancy was associated with a higher risk of NHL in children. Elevated odds were found for T-cell and B-cell lymphomas in both young and older children [64]. These finds were confirmed in a different study which demonstrated an excessive risk of lymphoma in children whose parents were exposed to pesticides during the preconception period [65].

#### 3.1.3. Brain Tumor

Research also demonstrates a link between parental exposure to pesticides and the increased risk of brain tumors in children. A case-control study of pediatric brain tumors conducted in Los Angeles, California investigated the association with household pesticides exposure from pregnancy to diagnosis. The risk of brain tumors was significantly elevated for prenatal exposure to flea/tick pesticides, particularly among children less than five years old at diagnosis [66]. These findings were confirmed by another study involving 526 one-to-one matched case-control pairs. It found an association between the risk of brain cancer, especially astrocytoma, in children under 10 years of age and parental exposure to pesticides during the two years prior to birth [67].

#### 3.1.4. Neuroblastoma

Neuroblastoma mostly affects children under the age of five and is the most common type of neoplasm in children under the age of 1. A recent meta-analysis suggests that pre-conception and prenatal parental exposure to pesticides is linked to increased rates of these pediatric tumors [68]. Overall, the associations between exposures to pesticides are stronger for children diagnosed after the age of one compared with infants in another study [69]. In agreement with these findings, in vitro studies showed that the treatment of neuroblast cells with organophosphates leads to the formation of abnormal neurite-like structures [70].

#### 3.1.5. Ewing Sarcoma and Wilms Tumor

The literature also links parental occupational exposure to pesticides with the development of Ewing Sarcoma in offspring. An Australian study linked continued exposure to pesticides, including exposure during conception and gestation, to a greater likelihood of Ewing Sarcoma development in the progeny, especially at younger ages (0–20 years of age) [47,71].

Consistent with that, another study demonstrated the occurrence of Wilms tumor in children whose parents were exposed to pesticides during the preconception period. However, the reported number for this type of tumor was significantly lower compared to the incidence of leukemia, lymphoma and brain tumors [65].

#### 3.1.6. Retinoblastoma

A case-control study linked parental exposure to pesticides to the emergence of retinoblastoma in children. The development of retinoblastoma is associated with inactivation of the RB1 gene. The RB1 gene inactivation occurs during DNA replication in proliferating retinal progenitor cells, and retinal progenitor cell proliferation occurs only in the fetal retina. Since genetic inactivation is required in both copies of RB1 to induce retinoblastoma, it has been proposed that prenatal exposure to pesticides could act as a second hit [72].

### 3.2. Adult Cancers

Epidemiologic analyses using the Child Health and Development Studies (CHDS) and other cohorts showed that developmental exposures to DDT is associated with increased rates of breast cancer [43,73,74]. For instance, daughters of women exposed to high levels of DDT in pregnancy have about a five-fold increase in breast cancer risk and are more likely to be diagnosed with advanced stage tumors [43].

Accordingly, Wu and colleagues [75] evaluated the association between prenatal exposure to DDT and DNA methylation, using blood samples from middle-age women, to better understand the mechanisms involved in the increased risk of breast cancer. They identified three genes associated to breast cancer susceptibility (CCDC85A, CYP1A1 and ZFPM2) with altered DNA methylation patterns. Their data suggest that prenatal exposure to DDT may have lifelong consequences, as it causes alterations in genes important for breast cancer.

Ongoing mouse studies in our lab show that preconception paternal exposure to DDT increases mammary tumorigenesis and tumor aggressiveness in female offspring. The offspring’s phenotype is linked to alterations in the sperm small non-coding RNA load of males exposed to DDT [19].

In addition to parental pesticide exposures, other endocrine disruptors have also been linked to adult cancers in the next generation. In rats, maternal exposure to endocrine disrupting chemicals during pregnancy increases the risk of breast cancer in three generations of offspring. The breast cancer phenotype in offspring was associated with DNA methylation changes in genes involved in stem cell specification in mammary tissues. These findings suggest that breast cancer is transmitted through epigenetic inheritance caused by intrauterine exposures and could explain a portion of familial breast cancer that are not explained by gene mutations [21].

The literature also demonstrates an association between in utero exposure to the endocrine disruptor diethylstilbestrol (DES) and breast cancer. DES is a synthetic estrogen that used to be prescribed to pregnant women to prevent miscarriages and premature births. However, serious adverse effects were later shown in the offspring resulting from its use, such as cancer [76]. The data show that the incidence of breast cancer is at least two-fold higher in daughters of mothers exposed to DES [77]. In line with that, in utero exposure to DES increased the expression of EZH2 and histone H3K27 trimethylation in normal and tumorigenic mammary cells [78].

A study conducted in the Netherlands followed 12,091 women exposed to DES in utero from 1992 to 2008. The data revealed the occurrence of 348 cases of cancer, with a median age at the end of the follow-up of 44 years. A significant increase in the risk of clear cell adenocarcinoma of the vagina and cervix (above 40 years of age) and an increased risk of melanoma (before 40 years of age) were observed [79] in DES exposed women.

Newbold and colleagues confirmed the intergenerational effect of DES in an animal model. They observed an increased susceptibility of reproductive tract tumors, including uterine adenocarcinomas, in daughters and granddaughters of females exposed to DES in pregnancy [80].

Table 1 provides a summary of pediatric and adult cancers associated with parental exposure to pesticides and/or endocrine disruptors.

## 4. Potential Mechanisms of Intergenerational and Transgenerational Transmission of Environmentally-Induced Cancer Susceptibility

### 4.1. Intergenerational versus Transgenerational Transmission

Environmentally-induced epigenetic alterations can be inherited through the parental germline and culminate in the development of diseases. However, it is important to differentiate between intergenerational and transgenerational inheritance. In intergenerational inheritance, an organism undergoes epigenetic changes resulting from an environmental insult. For example, in utero exposures could affect the developing fetus or its germ line, and these changes could be passed to its immediate descendants. In transgenerational inheritance, on the other hand, phenotypic changes are found in subsequent generations that were not directly exposed to the initial insult. In other words, inherited traits are passed on and maintained between generations, even in the absence of the stimulating factor [81].

It is important to consider that there are barriers to transgenerational epigenetic inheritance. In mammals, epigenetic reprogramming occurs in the zygote immediately after fertilization and during germ cell specification [81]. In both cases epigenetic patterns are reset and then re-established and is not entirely clear whether environmentally-induced epigenetic changes could escape this process.

### 4.2. Epigenetic Carriers

Epigenetics can be defined as the study of changes in gene functions that are mitotically and/or meiotically inherited and that does not lead to a change in the DNA sequence [82]. There are three main carriers of epigenetic information: DNA methylation, histone modifications and non-coding RNAs [83]. These epigenetic mechanisms control gene expression in somatic cells. However, it is becoming clear that epigenetic information can function as molecular memory of past environmental exposures and be passed from one generation to another via the germline [84]. In this section, we discuss how these epigenetic mechanisms can play a potential role in the intergenerational and transgenerational transmission of disease predisposition (Figure 1), including cancer susceptibility in the context of pesticide and other environmental exposures.

#### 4.2.1. DNA Methylation

One of the main epigenetic markers is DNA methylation, the covalent addition of a methyl group to a DNA nucleotide. Heritable cytosine methylation primarily occurs in the context of the symmetric CpG (5′-Cytosinephosphate-Guanine-3′) dinucleotide and the replication results in two daughter genomes each carrying a hemi-methylated CpG, providing a substrate for the enzyme DNA methyltransferase (DNMT)1 [85,86].

DNA methylation patterns are first established during embryonic development. This de novo DNA methylation occurs during cellular differentiation or germ cell and zygotic reprogramming [87]. De novo methylation is catalyzed by DNA methyltransferases such as DNMT3a, DNMT3b, and DNMT3L to establish methylation on unmodified DNA [88]. Once established, DNA methylation patterns of one strand are copied onto the daughter strand during replication by DNMT1.

The mammalian genome is generally CpG poor, with the exception of CpG islands (CGIs), which are relatively small genomic regions [89]. The sperm genome is heavily methylated, but also has focal hypomethylation in the CpG islands. Methylation in the oocyte exhibits far greater variance across the genome, as hypomethylated DNA is found in highly transcribed genes [85]. However, for transgenerational inheritance of DNA methylation to occur, some part of the genome would have to evade the pre-fertilization reprogramming [90].

#### 4.2.2. Histone Modifications

A histone protein octamer forms the core of the nucleosome, the structural unit of chromatin around which DNA is wrapped. Post-translational modifications, such methylation and acetylation, in histone tails determine the shape of the nucleosomes and change how DNA interacts with those proteins. Those histone tail modifications can either strengthen or weaken the amino-acid bonds between the histone and the DNA and affect whether a genomic region is accessible to the transcriptional machinery. In addition to its importance in the regulation of gene expression, many histone modifications have been implicated in transgenerational epigenetic inheritance. The mechanisms related to histone modifications involved in transgenerational epigenetics are mainly the repressive marks histone H3 lysine 9 trimethylation (H3K9me3) and H3 lysine 27 trimethylation (H3K27me3) [86,87].

In mammals, histones in both the female and male germline undergo extensive reprogramming resulting in a chromatin markedly different from that present in somatic cells. The chromatin in both the sperm and the oocyte undergoes widespread histone replacement, and the resulting chromatin is characterized by germline-specific histone variants. Histone modifications exhibit germline-specific patterns, so that methylation of H3K9 throughout the chromosome is related to sex chromosome inactivation [85,87].

In contrast to DNA methylation, histones and their methylation marks are highly conserved across species, making them attractive candidates for conserved mechanism for intergenerational inheritance. The sperm chromatin landscape undergoes a pronounced change during germ cell maturation, however chromatin remodeling during spermatogenesis results in a nucleus smaller than in an interphase somatic cell, and makes it difficult to establish the genomic profile of histone variants and changes in both sperm and oocyte [87].

#### 4.2.3. Non-Coding RNAs

Small non-coding RNAs regulate gene expression at both the transcriptional and posttranscriptional level. This class of RNAs include small interfering RNAs (siRNAs), Piwi-interacting RNAs (piRNAs), microRNAs (miRNAs) and tRNA-derived small RNAs (tsRNAs). Small RNA can silence gene transcription by altering the epigenetic state of a gene (e.g., piRNAs and siRNAs) or through post-transcriptional regulation, which leads to mRNA degradation (e.g., miRNAs). In both cases, they silence gene activity independently of DNA sequence. But in animals, the most common RNA-based mechanisms involved in transgenerational epigenetic inheritance are the small RNA silencing pathways [86].

Although piRNAs are the main RNA pathway in germline development, mature germ cells carry additional, or entirely distinct, RNA loads. In mammals, piRNAs are almost completely absent from mature sperm. Instead, its cargo is comprised primarily of tsRNAs and to a lesser extent miRNAs. Environmentally-induced epigenetic reprogramming events occur during post-testicular maturation in the epididymis and alter the composition of the sperm RNA load. These small RNAs produced in epididymal epithelial cells appear to be trafficked to maturing sperm in exosomes [85,91].

Studies show that small non-coding RNAs, which are abundant in sperm [22,92], play a critical role [91,93,94] in epigenetic inheritance. Several recent published reports demonstrated that the sperm RNA load can transmit environmentally-induced phenotypes from fathers to offspring [18,22,91,93]. Some of those studies implicated specific classes of small RNAs. For instance, miRNAs overexpressed in the sperm of males exposed to different factors can replicate the effect of specific paternal exposures in offspring when injected into normal embryos [22,94,95]. Other studies have implicated tsRNAs [93]. Sperm small RNAs such miRNAs are delivered to the oocyte and modulate the transcriptome during the first few cell divisions, setting a signaling cascade that can impact embryonic development [92,96].

#### 4.2.4. Other Carriers of Epigenetic Information

DNA methylation patterns are erased and then re-established during embryonic development. Thus, environmentally-induced epigenetic alterations would need to survive this large-scale erasure in order to be a viable carrier of transgenerational epigenetic inheritance. However, emerging data suggests that epigenetic changes lost in early development can be reconstructed by secondary signals [86] such as transcriptional factors. Consistent with that, a recent study showed that demethylated regions in germ cells and early embryos are protected from DNA re-methylation by the biding of transcription factors [97]. Since many endocrine disruptors bind to ERα (estrogen receptor) and/or AR (androgen receptor), if they are present during the reprogramming of the germ line when the DNA is demethylated, it is possible that they could influence the recruitment patterns of those transcription factors. If they remain bound after fertilization, these transcriptions factors could affect cell differentiation during development and elicit the phenotypes observed after chemical exposure. The maintenance of these new transcription factors sites through several generations suggests the intriguing possibility that these alterations may be responsible for the phenotypic effects observed by initial exposure to endocrine disrupting chemicals but expressed generations later in their absence [98,99].

### 4.3. Pesticides, Endocrine Disruptors and Epigenetic Changes in the Germ-Line

Environmental factors can trigger epigenetic changes, and exposure to heavy metals, persistent organic pollutants, and other endocrine disrupting chemicals can act to modulate epigenetic carriers. Epigenetic alterations are capable of mediating specific mechanisms of toxicity upon exposure to chemical agents, and these modifications may remain even with the cessation of the exposure [100].

Studies in animal models and human cohorts have linked exposure to pesticides and other environmental toxicants to poor sperm quality. For instance, in utero exposure to DDE, a DDT metabolite, impairs male fertility and causes epigenetic alterations in the male germ-line in rats [9]. Other pesticides have similar effects: prenatal dioxin induces sperm epimutations and disease in subsequent generations [10]. Studies in human cohorts support the findings in animal models and show an association between pesticide exposure, including DDT, and sperm quality (reviewed in [11]). Our own data from a mouse model show that paternal exposure to DDT alters the small non-coding RNA load, particularly miRNAs, in sperm [19]. Exposure to mancozeb, a fungicide with wide agricultural and industrial use, is associated with epigenetic changes in the female germline and abnormal reproductive outcomes [12]. Accordingly, epidemiologic studies have linked paternal occupation—and associated pesticide exposure—to cancer and other diseases in their children [13,14,15,16] as discussed in Section 3.

Anway et al. [101] demonstrated that parental pesticide exposure was capable of altering epi-phenotypes in several generations. The transient exposure of a female rat to the pesticides vinclozolin or methoxychlor, during the period of gonadal sex determination, induced an adult phenotype in the F1 generation, with reduced spermatogenic capacity, increasing male infertility. These effects were transmitted up to four generations. In line with that, exposure to of rats dams to DDT during pregnancy promotes transgenerational inheritance of sperm differential DNA methylation regions in male offspring. The majority of the DMRs identified in the caudal sperm originated during the spermatogonia stages in the testis. The authors conclude that a cascade of epigenetic changes initiated in primordial germ cells is necessary to alter epigenetic programming during spermatogenesis to obtain the sperm epigenetics involved in the transgenerational epigenetic inheritance [102].

Another study by the same group [103] demonstrated that maternal exposure to a mixture of pesticides (permethrin and N,N-diethyl-meta-toluamide) led to exhibited 363 differential DNA methylation regions (DMR) in the F3 generation. This study also found that pubertal abnormalities, testis disease, and ovarian disease (primordial follicle loss and polycystic ovarian disease) were increased in F3 generation animals.

While the data on germline epigenetic effects of pesticides in humans is lacking, a study showed that exposure to low doses of POPs, mainly pesticides, was associated with global DNA hypomethylation in peripheral blood, estimated by the percent 5-methyl-cytosine (%5-mC) in Alu and LINE-1 assays, in healthy Koreans [100,104].

The association between endocrine disruptors and DNA methylation suggest that it plays an important role in transgenerational epigenetic inheritance of germline epimutations [103,105]. However, recent studies suggest that non-coding RNAs also play an important role in epigenetic inheritance [85,91]. In line with that, changes in non-coding RNAs were identified, in male offspring of rat dams exposed to DDT in pregnancy [106]. Consistent with those findings our data suggest that pre-conception paternal exposure to DDT, alter the non-coding RNA load in a mouse model [19].

Additionally, studies show that pesticides effects can be mediated by histone modifications. These histone modifications were mainly characterized by alterations in the acetylation status, altering the access of transcription factors to specific DNA sequences [100,107].

It has also been shown that environmental factors modulate epigenetic modifiers expression [108]. Consistent with that, genetic alterations in epigenetic modifiers have been linked to epigenetic inheritance. Kdm6a deletion in the male germ line of mice causes phenotypic effects on the offspring, with the descendants of the knocked out male Kdm6a show a higher incidence of tumors. These effects were observed in successive generations and were associated with hypermethylated regions in enhancers may alter regulation of genes involved in cancer initiation or progression. Epigenetic changes in male gametes may therefore impact cancer susceptibility in adult offspring [109].

The study of epigenetic inheritance in humans is limited, though evidence exists. An example is individuals exposed to prenatal hunger during the Dutch Hunger Winter (1944–1945). Studies have demonstrated that those individuals have increased susceptibility to chronic diseases such as diabetes, hypertension and cancer. This increased disease susceptibility is associated with DNA hypomethylation of the imprinted IGF2 gene compared with their unexposed, same-sex siblings, after 60 years [110,111]. Another recent historical cohort study showed that the paternal grandfather’s nutrition leads to a transgenerational increase in cancer mortality rates in grandsons [112]. In line with that, prenatal exposure to pesticides and other endocrines disruptors linked to intergenerational development of cancer have been shown to alter epigenetic marks, particularly DNA methylation, in the progeny [75,113]. A few studies have also demonstrated that environmental insults result in molecular changes in the human germline. These studies are usually conducted in mature sperm cells, as they are easier to access than oocytes. One example is the effect of obesity on epigenetic variation in human sperm. Epigenetic profiling of sperm from lean and obese men showed that obesity leads to distinct patterns of DNA methylation and non-coding RNAs, some of which are reversed by the loss of body weight [114]. Together, these data demonstrate that, in humans, environmental factors can have epigenetic effects in the parental germline and in the resulting offspring.

## 5. Conclusions and Challenges Ahead

The research summarized in this review provides an overview of the association between parental exposure to pesticides and endocrine disruptors and offspring’s cancer development both in human cohorts and animal models. We also discussed the possible mechanism by which environmentally-induced cancer predisposition could be transmitted between generations. We examined how exposures to toxicants in early life and in the preconception period can lead to germline epigenetic changes and their possible contribution to the propagation of environmentally-induced diseases over generations.

A better understanding of the mechanisms involved in environmentally-induced intergenerational and transgenerational transmission of cancer susceptibility, as well as which parental exposures most jeopardize the health and development of the child, could lead to preventive and therapeutic strategies.

Developing population-based preventive strategies to reduce intergenerational cancer incidence associated with pesticides and endocrine disruptors will not be an easy task. Performing population studies to identify epigenetic markers of ancestral exposure will likely take decades, given the temporal gap between the parental exposure and cancer onset in adult descendants. However, a focus on pediatric cancers (particularly those occurring in young children) could make this process more efficient. One strategy would be to conduct prospective studies in cohorts of individuals of reproductive age and their children.

Reducing intergenerational environmentally-induced epigenetically predisposition to cancers and other diseases would require interventions in both in the pre-conception and prenatal windows. As discussed in this review, environmental insults alter non-coding RNAs and other epigenetic marks in the male germline that could potentially be used a as maker of exposure in humans. However, the implementation of such strategies in the clinical setting would require specific medical recommendations not only to women undergoing pre-conception counseling, but also to men.

Finally, there is a need for public health policies that control the use of and exposure to pesticides, especially in disadvantaged communities and in developing countries. It is essential to monitor and raise awareness about the exposure to these chemical agents to minimize the occurrence of damage to the health of both the directly exposed individual and their descendants.

## Figures and Tables

**Figure 1 ijms-23-04671-f001:**
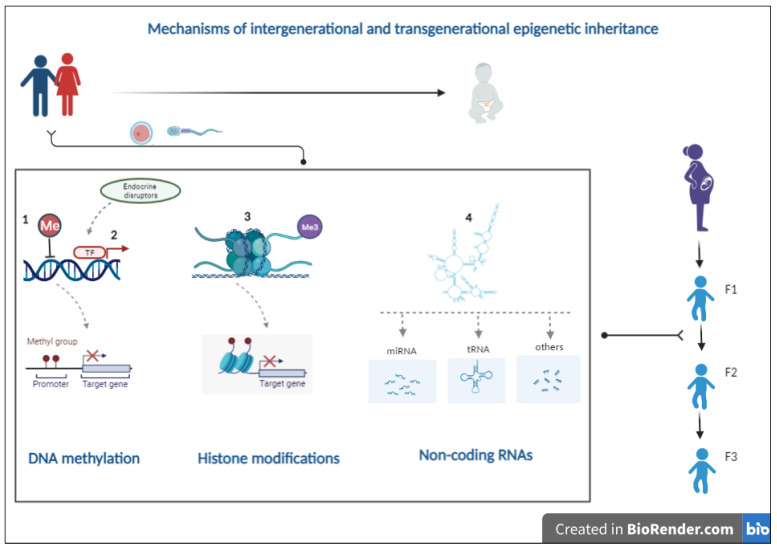
Parental pre-conception and in utero exposure to chemical substances, such as pesticides, cause epigenetic alterations in the germline that can be transmitted between generations and affect disease (including cancer) susceptibility in the progeny. Several mechanisms play a potential role in intergenerational and transgenerational transmission of disease predisposition including DNA methylation (1), transcription factor-associated DNA methylation patterns (2), histone modifications (3) and non-coding RNAs (4). Details are described in Section 4.

**Table 1 ijms-23-04671-t001:** Pediatric and adult cancers resulting from parental exposure to pesticides or endocrine disruptors.

	Type of Cancer	Pesticide or EDC	Reference
**Pediatric**			
	Leukemia	Organophosphates Propoxur; Cypermethrin; Chlorpyrifos	[14,48,58,59,60,61,62,63]
	Hodgkin and Non-Hodgkin’s Lymphoma	Occupational pesticide exposure	[64,65]
	Brain tumor	Organochlorine; Methyl bromide	[66,67]
	Neuroblastoma	Residential pesticides, Iazinon, Glyphosate, Malathion, Parathion, and Tetrachlorvinphos	[68,69,70]
	Ewing Sarcoma and Wilms tumor	Occupational pesticide exposure	[47,65,71]
	Retinoblastoma	Residential pesticides	[72]
**Adult**			
	Breast cancer	DDT	[19,21,43,73,74,75]
	Cell adenocarcinoma of the vagina and cervix	DES	[79]
	Melanoma	DES	[79]
	Uterine adenocarcinoma	DES	[80]

**EDC**, endocrine disrupting chemical; **DDT**, dichlorodiphenyltrichloroethane; **DES**, diethylstilbestrol.

## Data Availability

Not applicable.

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
