# Peer review of "Epigenetic Inheritance: Intergenerational Effects of Pesticides and Other Endocrine Disruptors on Cancer Development"

_ijms, 2022, doi:10.3390/ijms23094671_

Round 1

Reviewer 1 Report

The aim of this manuscript is to explore scientific evidence on parental exposure to pesiticides and endocrine disruptors, by evaluating the prediposition to cancer, in the next generation, both in human and animal studies.

This review shows rich content, providing a deep insight for some works: I found it to be well-written and accessible, providing sufficient information for the non-expert while also achieving a balance of detail for those with more expertise in the field. This is the additional point, which makes this manuscript original, in comparison to published literature. Even if the manuscript provides an organic overview, with a densely organized structure and based on well-synthetized evidence, there are aspects to be mentioned, to make the article fully readable. For these reasons, the manuscript requires minor changes.

Please find below an enumerated list of comments on my review of the manuscript:

INTRODUCTION:

LINE 38-39: Endrocrine disruptors can cause adverse health effects, with a universal approach: in this context, several balances are damaged by environmental exposure to this compounds, specifically significant effects are asscoiated to male and female reproduction, breast development, neuroendocrinology and obesity, as suggested by several and recent studies (see, for reference: Encarnação T, Pais AA, Campos MG, Burrows HD. Endocrine disrupting chemicals: Impact on human health, wildlife and the environment. Sci Prog. 2019 Mar;102(1):3-42. doi: 10.1177/0036850419826802. Epub 2019 Jan 1. PMID: 31829784). In this context, the authors should introduce this aspect, in order to provide to the readers recent evidence on the pleiotropic and dangerous effects of endocrine disruptors compounds in the overall human health.

LINE 56: In this context, female fertility and reproductive health are sensitive to toxic exposure, specifically to endocrine disruptors pollutans, which exerts long-term adverse effects. For example, environmental exposure to pesticides and endocrine disruptor compounds may influence the full developmental competence of the female germ cell, by enhancing specific histone modifications, as suggested by several and recent studies (see, for reference: Bianchi, S.; Nottola, S.A.; Torge, D.; Palmerini, M.G.; Necozione, S.; Macchiarelli, G. Association between Female Reproductive Health and Mancozeb: Systematic Review of Experimental Models. Int. J. Environ. Res. Public Health 202017, 2580. https://doi.org/10.3390/ijerph17072580). Due to their persistence and versatile profile, pesticides are considered an epigenetic hazard and powerful pollutants, which interact with female/male reproductive phenotypes, changing directly or indirectly the inner molecular and cellular balances.

The conclusion of this manuscript is perfectly in line with the main purpouse of the paper: the authors have designed and conducted the study properly and they also discuss the limitation of this work, key point for future research. In this cotext, conclusions are well written and present an adequate balance between the description of previous findings and the results presented by the authors. Finally, this manuscript also presents a basic structure, properly divided and characterized by organic and detailed figures and tables.In conclusion, this manuscript is densely presented and well organized, based on well-synthetized evidences. The authors were lucid in their style of writing, making it easy to read and understand the message, portrayed in the manuscript. Besides, the methodology design was rigorous and appropriately implemented within the study. However, many of the topics are very concisely covered. This manuscript provided a comprehensive review of current knowledge in this field. Moreover, this research have futuristic importance and could be potential for future research. However, the minor concern of this manuscript is with the introductive section: for these reasons, I have minor comments only for the introductive section, for improvement before acceptance for publication. The article is accurate and provides relevant information on the topic and I suggest minor changes to be made in order to maximize its scientific impact. I would accept this manuscript, if the comments are addressed properly.

Author Response

Answers to Reviewers’ Comments

We thank reviewers for their insightful and constructive comments on our review article. We have addressed each comment point-by-point below (in blue font). Any modifications in the original manuscript text in response to reviewers’ critiques are marked by highlighted blue font.

REVIEWER 1

Comments and Suggestions for Authors: The aim of this manuscript is to explore scientific evidence on parental exposure to pesiticides and endocrine disruptors, by evaluating the prediposition to cancer, in the next generation, both in human and animal studies.

This review shows rich content, providing a deep insight for some works: I found it to be well-written and accessible, providing sufficient information for the non-expert while also achieving a balance of detail for those with more expertise in the field. This is the additional point, which makes this manuscript original, in comparison to published literature. Even if the manuscript provides an organic overview, with a densely organized structure and based on well-synthetized evidence, there are aspects to be mentioned, to make the article fully readable. For these reasons, the manuscript requires minor changes.

Response: We thank reviewer 1 for their comments. We have addressed the minor concerns raised. See specific answers below.

INTRODUCTION:

  • LINE 38-39: Endocrine disruptors can cause adverse health effects, with a universal approach: in this context, several balances are damaged by environmental exposure to this compounds, specifically significant effects are asscoiated to male and female reproduction, breast development, neuroendocrinology and obesity, as suggested by several and recent studies (see, for reference: Encarnação T, Pais AA, Campos MG, Burrows HD. Endocrine disrupting chemicals: Impact on human health, wildlife and the environment. Sci Prog. 2019 Mar;102(1):3-42. doi: 10.1177/0036850419826802. Epub 2019 Jan 1. PMID: 31829784). In this context, the authors should introduce this aspect, in order to provide to the readers recent evidence on the pleiotropic and dangerous effects of endocrine disruptors compounds in the overall human health.

Response: This is an important point and additional information on the pleiotropic health effect of environmental toxicants has been added to the introduction.

  • LINE 56: In this context, female fertility and reproductive health are sensitive to toxic exposure, specifically to endocrine disruptors pollutans, which exerts long-term adverse effects. For example, environmental exposure to pesticides and endocrine disruptor compounds may influence the full developmental competence of the female germ cell, by enhancing specific histone modifications, as suggested by several and recent studies (see, for reference: Bianchi, S.; Nottola, S.A.; Torge, D.; Palmerini, M.G.; Necozione, S.; Macchiarelli, G. Association between Female Reproductive Health and Mancozeb: Systematic Review of Experimental Models. Int. J. Environ. Res. Public Health 2020, 17, 2580. https://doi.org/10.3390/ijerph17072580). Due to their persistence and versatile profile, pesticides are considered an epigenetic hazard and powerful pollutants, which interact with female/male reproductive phenotypes, changing directly or indirectly the inner molecular and cellular balances.

Response: We thank the reviewer for bringing the literature on the specific effects of environmental toxins on the female germline to our attention. Additional information focusing on the female germline has been has been added to the introduction as suggested as well as in section 4.3.

  • The conclusion of this manuscript is perfectly in line with the main purpouse of the paper: the authors have designed and conducted the study properly and they also discuss the limitation of this work, key point for future research. In this cotext, conclusions are well written and present an adequate balance between the description of previous findings and the results presented by the authors. Finally, this manuscript also presents a basic structure, properly divided and characterized by organic and detailed figures and tables.In conclusion, this manuscript is densely presented and well organized, based on well-synthetized evidences. The authors were lucid in their style of writing, making it easy to read and understand the message, portrayed in the manuscript. Besides, the methodology design was rigorous and appropriately implemented within the study. However, many of the topics are very concisely covered. This manuscript provided a comprehensive review of current knowledge in this field. Moreover, this research have futuristic importance and could be potential for future research. However, the minor concern of this manuscript is with the introductive section: for these reasons, I have minor comments only for the introductive section, for improvement before acceptance for publication. The article is accurate and provides relevant information on the topic and I suggest minor changes to be made in order to maximize its scientific impact. I would accept this manuscript, if the comments are addressed properly.

Response: We are grateful to reviewer 1 for their appreciation of our manuscript’s content. We have addressed the concerns raised and modified the manuscript text accordingly.

Reviewer 2 Report

This review attempted to summarize the evidence that parental exposure to pesticides increases the risk of cancer in the offspring and next generations. The authors describe epigenetics as a basis for inter and transgenerational transmission of parental exposure. Unfortunately, there are several shortcomings to this review.

Major points: Some parts look like a description of multiple articles without discussion.

Section 4.3 needs to be much more documented. The review should provide a more comprehensive description of the molecular mechanisms involved in tumorigenesis when they are available. For instance, the role of Line and Alu in cancer should be explained.

The genes controlled by epigenetics and involved in carcinogenesis after exposure to pesticides should be cited. The organs affected should be precised.

There are discrepancies between the title and the content of the review. According to the title, pesticides are the subject of the review. However, in the manuscript, other toxicants are also described. The title should be changed or the review should be focused on pesticides only, with or without endocrine disrupting activity.

The part describing the mechanisms involved in the transgenerational effects of the Dutch winter hunger should better highlight the potential of epigenetic transmission in the context of exposure to pesticides.

Minor points:

The words “also”, “other”, “another” appear very often in the text, which gives the impression of a succession of descriptions.

This area of research has been already covered partly by other groups, e.g. “Sperm epigenome as a marker of environmental exposure and lifestyle, at the origin of diseases inheritance, Siddeek B, et al., Mutat Res Rev Mutat Res. 2018”, “Tumour predisposition and cancer syndromes as models to study gene-environment interactions, Carbone M et al, Nat Rev Cancer. 2020”. The authors could cite these works and shorten the part 4.2.

Table 1 should indicate which pesticides have been studied.

In the title: remove “the” before “effects”.

Line 475: add “of” between “effect” and “obesity”.

Author Response

Answers to Reviewers’ Comments

We thank reviewers for their insightful and constructive comments on our review article. We have addressed each comment point-by-point below (in blue font). Any modifications in the original manuscript text in response to reviewers’ critiques are marked by highlighted blue font

-

REVIWER 2

Comments and Suggestions for Authors

This review attempted to summarize the evidence that parental exposure to pesticides increases the risk of cancer in the offspring and next generations. The authors describe epigenetics as a basis for inter and transgenerational transmission of parental exposure. Unfortunately, there are several shortcomings to this review.

Response: We thank reviewer 2 for constructive critiques that helped us improve our manuscript.  Below, we address major and minor concerns raised and modified the manuscript text accordingly.

Major points:

  • Some parts look like a description of multiple articles without discussion.

Response: We appreciate the reviewer’s comment and have revised the manuscript to ensure that different part of the article is more connected to each other. Please see specific answers below.

  • Section 4.3 needs to be much more documented. The review should provide a more comprehensive description of the molecular mechanisms involved in tumorigenesis when they are available…. The genes controlled by epigenetics and involved in carcinogenesis after exposure to pesticides should be cited.

Response: We thank the reviewer for their comment. We have introduced more discussion on the potential molecular and epigenetic mechanisms behind environmentally-induced intergenerational tumorigenesis where the evidence is available throughout section 3 (see both section 3.1 and 3.2). The reason why this information has been added to section 3 instead of section 4.3, as suggested by the reviewer, is that section 4.3. is titled “Pesticides, endocrine disruptors and epigenetic changes in the germ-line’. Therefore, in section 4.3 we reviewed the evidence regarding the effects of pesticides and endocrine disruptors on the germline, while we discuss the mechanisms involved in tumorigenesis more specifically in section 3.

  • There are discrepancies between the title and the content of the review. According to the title, pesticides are the subject of the review. However, in the manuscript, other toxicants are also described. The title should be changed or the review should be focused on pesticides only, with or without endocrine disrupting activity.

Response: We modified the title as requested by the reviewer:

New title: Epigenetic inheritance: Intergenerational effects of pesticides and other endocrine disruptors on cancer development

  • The part describing the mechanisms involved in the transgenerational effects of the Dutch winter hunger should better highlight the potential of epigenetic transmission in the context of exposure to pesticides.

Response: We have added additional information highlighting the potential of epigenetic transmission in the context of exposure to pesticides and other endocrine disruptors in that section.

Minor points:

  • The words “also”, “other”, “another” appear very often in the text, which gives the impression of a succession of descriptions.

Response: We have edited the manuscript to make it a more streamlined reading experience.

  • This area of research has been already covered partly by other groups, e.g. “Sperm epigenome as a marker of environmental exposure and lifestyle, at the origin of diseases inheritance, Siddeek B, et al., Mutat Res Rev Mutat Res. 2018”, “Tumour predisposition and cancer syndromes as models to study gene-environment interactions, Carbone M et al, Nat Rev Cancer. 2020”.

Response: We thank the reviewer for bringing these articles to our attention. The references suggested by the reviewer are now cited in sections 3 and 4.2.

  • Table 1 should indicate which pesticides have been studied.

Response: We appreciate the reviewer’s suggestion.  Many of the epidemiologic studies we cite in our review assess pesticides as category or exposure assumed by occupational or residential exposure of subjects. In some cases, exposure to specific pesticides was assessed or measured in circulation of participants (e.g. Cohn et al, 2015).  Thus, based on information we could extract from the literature, examples of pesticides related to cancer types were added to the table 1.

  • In the title: remove “the” before “effects”.

Response: We have corrected the title.

  • Line 475: add “of” between “effect” and “obesity”.

Response: We have added the missing word.

Round 2

Reviewer 2 Report

The authors have addressed my concerns.